# Glycogen Storage Disease Type Ia: Current Management Options, Burden and Unmet Needs

**DOI:** 10.3390/nu13113828

**Published:** 2021-10-27

**Authors:** Terry G. J. Derks, David F. Rodriguez-Buritica, Ayesha Ahmad, Foekje de Boer, María L. Couce, Sarah C. Grünert, Philippe Labrune, Nerea López Maldonado, Carolina Fischinger Moura de Souza, Rebecca Riba-Wolman, Alessandro Rossi, Heather Saavedra, Rupal Naik Gupta, Vassili Valayannopoulos, John Mitchell

**Affiliations:** 1Division of Metabolic Diseases, Beatrix Children’s Hospital, University of Groningen, University Medical Center Groningen, 9700 RB Groningen, The Netherlands; f.de.boer@umcg.nl (F.d.B.); a.rossi@umcg.nl (A.R.); 2Department of Pediatrics, Division of Medical Genetics, McGovern Medical School at the University of Texas Health Science Center at Houston (UTHealth Houston) and Children’s Memorial Hermann Hospital, Houston, TX 77030, USA; david.f.rodriguezburitica@uth.tmc.edu (D.F.R.-B.); heather.saavedra@uth.tmc.edu (H.S.); 3Department of Pediatrics, Division of Pediatric Genetics, Metabolism and Genomic Medicine, University of Michigan, Ann Arbor, MI 48109, USA; ayeshaah@med.umich.edu; 4IDIS, CIBERER, MetabERN, University Clinical Hospital of Santiago de Compostela, 15706 Santiago de Compostela, Spain; maria.luz.couce.pico@sergas.es; 5Department of General Pediatrics, Adolescent Medicine and Neonatology, Faculty of Medicine, Medical Center-University of Freiburg, 79106 Freiburg, Germany; sarah.gruenert@uniklinik-freiburg.de; 6APHP, Université Paris-Saclay, Hôpital Antoine-Béclère, 92140 Clamart, France; philippe.labrune@aphp.fr; 7Inserm U 1195, Paris-Saclay University, 94276 Le Kremlin Bicêtre, France; 8Piera Health Center, Catalan Institute of Health, 08007 Barcelona, Spain; nlopezm.cc.ics@gencat.cat; 9Autonomous University of Barcelona, 08193 Barcelona, Spain; 10Medical Genetics Service, HCPA, Porto Alegre 90035-903, Brazil; cfsouza@hcpa.edu.br; 11Connecticut Children’s Medical Center, Department of Pediatrics, Division of Endocrinology, University of Connecticut, Farmington, CT 06032, USA; rriba@connecticutchildrens.org; 12Department of Translational Medicine, Section of Paediatrics, University of Naples “Federico II”, 80131 Naples, Italy; 13Ultragenyx Pharmaceutical Inc., Novato, CA 94949, USA; rgupta@ultragenyx.com (R.N.G.); vvalayannopoulos@ultragenyx.com (V.V.); 14Department of Pediatrics, Division of Pediatric Endocrinology, Montreal Children’s Hospital, McGill University Health Center, Montreal, QC H4A 3J1, Canada; john.james.michell.med@ssss.gouv.qc.ca

**Keywords:** glycogen storage disease type Ia, dietary treatment, uncooked cornstarch, burden of disease, unmet need, long-term complications, quality of life

## Abstract

Glycogen storage disease type Ia (GSDIa) is caused by defective glucose-6-phosphatase, a key enzyme in carbohydrate metabolism. Affected individuals cannot release glucose during fasting and accumulate excess glycogen and fat in the liver and kidney, putting them at risk of severe hypoglycaemia and secondary metabolic perturbations. Good glycaemic/metabolic control through strict dietary treatment and regular doses of uncooked cornstarch (UCCS) is essential for preventing hypoglycaemia and long-term complications. Dietary treatment has improved the prognosis for patients with GSDIa; however, the disease itself, its management and monitoring have significant physical, psychological and psychosocial burden on individuals and parents/caregivers. Hypoglycaemia risk persists if a single dose of UCCS is delayed/missed or in cases of gastrointestinal intolerance. UCCS therapy is imprecise, does not treat the cause of disease, may trigger secondary metabolic manifestations and may not prevent long-term complications. We review the importance of and challenges associated with achieving good glycaemic/metabolic control in individuals with GSDIa and how this should be balanced with age-specific psychosocial development towards independence, management of anxiety and preservation of quality of life (QoL). The unmet need for treatment strategies that address the cause of disease, restore glucose homeostasis, reduce the risk of hypoglycaemia/secondary metabolic perturbations and improve QoL is also discussed.

## 1. Introduction

Glycogen storage diseases (GSD) are a collection of inherited metabolic disorders caused by pathogenic variants in the genes that encode proteins involved in glycogen synthesis, glycogenolysis and/or gluconeogenesis. Glycogen storage disease type Ia (GSDIa; OMIM#232200), also known as von Gierke disease, is an inborn error of carbohydrate metabolism caused by bi-allelic pathogenic variants in the glucose-6-phosphatase gene (*G6PC*; OMIM*613742), which accounts for 80% of cases of type I glycogen storage disease [1]. Glucose-6-phosphatase (G6Pase) is selectively expressed in the liver, kidney and small intestine, where it functions in the endoplasmic reticulum to convert glucose-6-phosphate (G6P) into free glucose, the final common step in gluconeogenesis and glycogenolysis [2]. Loss of G6Pase function thereby impairs carbohydrate homeostasis during short-term fasting, putting individuals with GSDIa at risk of severe, life-threatening hypoglycaemic events over the course of their lifetime. Since the first case was described in the medical literature in 1929, there has been considerable basic, translational and clinical research into the disease. Dietary treatment has been the cornerstone of disease management since the introduction of continuous glucose therapy by continuous enteral feeding and consumption of uncooked cornstarch (UCCS) in the 1970s and 1980s, respectively [3,4,5]. In this review, we summarise the epidemiology, pathophysiology and clinical manifestations of GSDIa as well as the challenges with disease management.

## 2. Epidemiology

GSDIa is an ultra-rare (i.e., has a prevalence of less than one in 50,000) [6,7] autosomal recessive, pan-ethnic disorder that has an overall prevalence of one in 100,000 [8], The estimated carrier rate in the general population is approximately 1:150 [9,10,11]. The disease may be more prevalent in people of Ashkenazi Jewish [11,12], Mexican-Hispanic and Japanese heritage [13] due to a founder effect. Population differences in disease epidemiology may be due to different heterozygous carriers of pathogenic *G6PC* variants, such as c.247C>T in Ashkenazi Jews, c.379_380dupTA in Mexican-Hispanic and c.648G>T in individuals of Japanese ancestry. The carrier frequency for the c.247C>T variant among Ashkenazi Jews has been reported to be as high as 1:71, which is predictive of a prevalence five times higher than for the general Caucasian population [1]. A more recent publication reporting the results following carrier screening of Ashkenazi Jews in the New York metropolitan area estimated the carrier frequency for the c.247C>T allele to be 1:63 [14].

## 3. Pathophysiology and Clinical Manifestations

Under normal physiological conditions, plasma glucose concentration is tightly regulated and maintained within a narrow window, with a relatively low plasma pool and relatively high endogenous production. Several hormones act to prevent hypoglycaemia (e.g., glucagon, growth hormone, epinephrine and cortisol), whereas only insulin and, indirectly, glucagon like-peptide-1 by enhancing insulin secretion act to lower glucose concentration. Their interplay ensures the regulation of the metabolic pathways involved in endogenous glucose production and carbohydrate homeostasis. Hepatic G6P represents the traffic intersection of glycogenesis, glycogenolysis, glycolysis, gluconeogenesis and pentose phosphate pathways. Hepatic *G6PC* expression is regulated by a range of hormones and nutrients; its expression is stimulated by glucocorticoids, cyclic adenosine monophosphate (cAMP), leptin, β3-adrenergic receptor agonists, glucose and fatty acids, whereas it is inhibited by tumour necrosis factor α, interleukin-6 and insulin [15]. The G6Pase active site is located on the luminal side of the endoplasmic reticulum (ER) [16]. The G6P transporter is responsible for moving G6P from the cytoplasm into the ER lumen; therefore, the G6P transporter/G6Pase complex is responsible for glucose production by catalysing the terminal step of both the glycogenolysis and gluconeogenesis pathways [16]. G6P accumulation, secondary to G6Pase deficiency, may contribute to some of the observed metabolic perturbations in patients with GSDIa, such as hyperlactataemia, hyperuricaemia, hyperlipidaemia (studied and reviewed in [17]) and imbalanced cortisol homeostasis [18,19].

Patients with GSDIa typically present clinically at a median age of 6 months (range 1 day–12 years) [20]. In young infants, initial symptoms due to hypoglycaemia are often aspecific (such as feeding difficulties or apnoea) but may include features associated with epinephrine release (e.g., autonomic system (neurogenic) symptoms, such as tremors, paleness and sweating), cerebral glycopenia (e.g., symptoms such as hunger, irritability, seizures, somnolence and coma), lactic acidosis (e.g., respiratory compensation by hyperventilation) and secondary systemic consequences (e.g., cyanosis), improving in the short-term after feedings [21]. Severe (generally hypoketotic) hypoglycaemic episodes relating to increased fasting time or intercurrent illness or disease may occasionally lead to coma and sudden infant death. Older infants may also present with failure to thrive and a protruding abdomen. Other common clinical presentations include doll-like faces, growth failure, delayed development and hypotrophic muscles [11,22]. Histologically, the G6Pase deficiency results in an accumulation of more fat than glycogen in the liver and kidneys, with both contributing to progressive hepatomegaly and nephromegaly, respectively [23]. The broad spectrum of symptoms may make it difficult to identify patients who do not fit the classic phenotype, especially those with an unusually prolonged fasting tolerance [1]. Other conditions with a clinical presentation similar to GSDIa include GSD III, VI and IX, diabetes mellitus and Niemann–Pick type B [13]. 

In addition to hypoglycaemic events, GSDIa is associated with numerous secondary metabolic perturbations [23]. The G6Pase deficiency means G6P leaves the ER and accumulates in the cytoplasm and is shunted into alternative metabolic pathways (Figure 1) [11,12]. Metabolic consequences of elevated G6P common at presentation are hypercholesterolaemia, hypertriglyceridaemia, hyperuricaemia and hyperlactataemia [11,20,22]. Xanthomas and pancreatitis can subsequently occur as a consequence of high lipid levels [22]. The relationship between hyperlipidaemia and cardiovascular disease is unclear in patients with type I GSD. Thickening of the carotid intima media has been observed, which is suggestive of an increased risk of vascular dysfunction; however, case series have revealed inconsistent observations relating to vascular dysfunction and markers of cardiovascular disease risk [22]. Secondary biochemical abnormalities, such as inhibition of carnitine palmitoyltransferase I due to increased malonyl coenzyme A, leads to decreased ketone production during fasting [1].

The European Study on Glycogen Storage Disease (GSD) Type I was published in 2002. This study retrospectively analysed 288 patients with GSD type I (of whom 231 had GSDIa) and identified hepatocellular adenomas (HCA) and progressive renal disease as major causes of mortality [20]. HCAs are a well-documented complication in individuals with GSDIa, occurring in 16–75% of patients and equally in both sexes [1]. HCAs are usually detected in the second or third decade of life (median age 14.8 years) and are typically diagnosed following a liver ultrasound, computerised tomography or magnetic resonance imaging (MRI) [1,24]. To screen for adenomas, ultrasonography of the abdomen should be performed every year for patients aged 0–10 years and every 6 months for those aged >10 years [25]. Once adenomas have been detected, liver imaging (e.g., ultrasonography, MRI) should be repeated every 6–12 months (or more frequently depending on clinical and laboratory findings) to look for evidence of increasing lesion size or poorly defined margins [24,25]. HCAs in GSDIa are small, non-encapsulated adenomas that produce excess hepcidin (which contributes to anaemia) and are thought to be a result of poor metabolic control [23,26]. The pathophysiological mechanism is not fully understood; however, hyperlipidaemia is associated with an increased risk of HCA development. A retrospective chart review performed for 117 patients with GSDIa analysed HCA progression among two groups: those with a 5-year mean triglyceride concentration ≤500 mg/dL and >500 mg/dL. A significant difference in progression to HCA was observed in the patients with plasma triglyceride levels >500 mg/dL compared with those ≤500 mg/dL [27]. HCA must be monitored closely during pregnancy, and strict adherence to dietary treatment is required due to the potential for pre-existing adenomas to increase in size in the increased oestrogen state [28,29].

Renal manifestations of GSDIa appear in early childhood but probably go undetected without specific diagnostic evaluation [24]. The first sign of renal dysfunction is glomerular hyperfiltration, which commonly develops in the teenage years, followed by microalbuminuria and later proteinuria [11,23]. Individuals with GSDIa may also develop hypocitraturia and hypercalciuria, increasing their risk of nephrocalcinosis and nephrolithiasis, which may be exacerbated in the event of poorly controlled hyperuricemia and lactic acidosis.

Gastrointestinal and haematological disturbances are common. Gastrointestinal symptoms may include abdominal bloating and diarrhoea, possibly related to dietary treatment. Inflammatory bowel disease is well characterised in individuals with GSDIb; however, it has been reported in a few patients with GSDIa (potentially associated with chronic use of UCCS leading to alternations in the microbiota of the gastrointestinal tract) and may be under-recognised [22,30]. A coagulation defect attributed to acquired platelet dysfunction with prolonged bleeding times, decreased platelet adhesiveness and abnormal aggregation has been described. Bleeding manifestations include epistaxis, easy bruising, menorrhagia and excessive bleeding during surgical procedures [24]. Anaemia is common, with prevalence ranging from 17% to 60% across different age groups [24]. Anaemia is often mild and secondary to iron deficiency in children but may be more severe in adults, particularly those with HCA [31]. Other causes of anaemia include chronic lactic acidosis, chronic kidney disease and suboptimal metabolic control [24].

Long-term consequences of GSDIa if left untreated include growth retardation, delayed puberty, gout, arterial and (rarely) pulmonary hypertension, osteoporosis or osteopenia [32], polycystic ovary syndrome, HCA, hepatocellular carcinoma (HCC), chronic renal disease and renal failure, neuropathy, and cognitive delays and epilepsy because of repeated or severe hypoglycaemic events [9,23,24]. Nutritional deficiencies (protein and vitamin B12, folic acid and vitamin D) are often observed in treated individuals because of the dietary restrictions and lack of appetite associated with daily UCCS supplementation to prevent hypoglycaemia; therefore, a complete multivitamin is recommended [24].

In addition to clinical ascertainment in infancy, the introduction of next-generation sequencing has identified adults with GSDIa who may exhibit mild fasting intolerance [33] or present with HCC [34], HCA [35] or acute pancreatitis [36]. Most patients with GSDIa are compound heterozygous, which makes it difficult to draw genotype–phenotype-related conclusions; however, interesting observations can be made and important lessons learned from patients who are homozygous for specific *G6PC* pathogenic variants. Individuals who are homozygous for the c.648G>T pathogenic splicing variant (common in those of Japanese ancestry) may be at increased risk of developing HCC. Those with this nonhelical *G6PC* variant that only partially inactivates G6Pase [37] exhibit a milder phenotype with respect to onset and severity of hypoglycaemic events, which may explain why they are not diagnosed until adulthood [38,39]. It is likely that the HCC susceptibility among individuals with this variant is partially explained by the duration of untreated intrahepatic metabolic perturbations associated with late diagnosis and start of dietary treatment. Individuals who are homozygous for the c379_380dupTA *G6PC* pathogenic variant also have a milder phenotype (although less so than observed in patients of Japanese ancestry) with increased residual G6Pase activity leading to longer fasting times. However, these patients display a higher rate of chronic liver complications, such as HCC, compared with patients who are compound heterozygous (unpublished clinical observations by Dr David Rodriguez-Buritica). Additionally, individuals with GSDIa who are homozygous for the c.562G>C *G6PC* pathogenic variant display neutropenia, neutrophil dysfunction and recurrent infections characteristic of GSDIb [40]. This phenotype is not observed for individuals who are compound heterozygous for the c.562G>C *G6PC* variant [13]. 

Dietary treatment and drug therapy for symptoms and secondary complications has improved survival and decreased some of the disease-associated morbidities; however, current disease management strategies are not without risk of failure and long-term prognosis remains uncertain given the number and variety of disease complications [19,20,41].

## 4. Current Strategies for Disease Management

Management of individuals with GSDIa involves a high degree of personalised medicine. The cornerstone is based on medically prescribed dietary treatment, and there are no approved pharmacological therapies that address or correct the underlying cause of disease [24,42]. Instead, oral glucose replacement therapy in the form of UCCS is used, in addition to regular meals and snacks, to manage blood glucose levels, and adjunct therapies are administered to manage secondary manifestations of the disease [22,24,25]. In addition to maintaining euglycaemia (thus preventing hypoglycaemia), the aims of disease management are to avoid acute metabolic derangement, prevent acute and long-term complications and ensure normal psychomotor development and good quality of life (QoL) [25]. The biomedical targets in GSDIa are summarised in Box 1. Additional therapies may be required to treat hyperuricaemia, hyperfiltration, albuminuria, hyperlipidaemia, hypocitraturia, arterial hypertension and osteoporosis or to prevent long-term complications, such as HCA, HCC, and chronic renal disease and renal failure, as well as to normalise growth [22,24,25].

Box 1Biomedical targets in Glycogen Storage Disease type Ia.1.Hypoglycaemia associated with the glycogenolysis defect.2.Hepatomegaly, adenomas, adenocarcinomas and chronic kidney disease associated with glycogen accumulation.3.Hypoglycaemia and hyperlactataemia associated with the gluconeogenesis defect.4.Dyslipidaemia (hypertriglyceridaemia), hyperuricaemia and hepatic steatosis associated with glucose-6-phosphate accumulation.5.Obesity, metabolic syndrome, anorexia and malnutrition associated with current disease management.

The first treatment of GSD in the 1960s involved portocaval transposition and aimed to reduce hepatic first-pass metabolism of dietary glucose [43]; however, the use of parenteral nutrition proved to be more effective, and the disease was effectively managed with continuous nocturnal gastric drip-feeding (CNGDF) to ensure prolonged carbohydrate intake [3,4]. Prior to the introduction of UCCS, infants are managed with sucrose-free soy infant formula or a formula that is free of sucrose, fructose and lactose; bolus feeds are typically required every 2 h [24]. UCCS has been used to improve maintenance of glucose concentrations between meals in patients with GSDIa since the 1980s and is typically introduced between 6 months and 1 year of age [5,24]. European guidelines published in 2002 recommend CNGDF starts in infancy and continues into adolescence [25]. More recent US guidelines also recommend overnight gastric feeding as an alternative to waking the infant frequently to monitor blood glucose and offer feedings. Parents and/or older children (from the age of 4 years) can be trained to insert a nasogastric tube (NGT), or a percutaneous endoscopic gastrostomy (PEG) may be performed to insert a PEG-/G-tube to ensure treatment can be provided in times of illness or refusal to eat [24]. Both NGT and PEG-/G-tubes are associated with complications that may affect the child’s and their family’s QoL; such complications may be more acute with NGT when they are associated with insertion of the tube itself [44]. PEG/G-tubes may not be suitable for all individuals with GSDIa due to the risk of infection. Similar to individuals with GSDIb, those with the c.562G>C *G6PC* pathogenic variant may be susceptible to neutropenia, and PEG/G-tubes should only be placed if granulocyte colony-stimulating factor (Neupogen) [24] or another treatment (e.g., empagliflozin) is being administered. Case reports describe how CNGDF is not without risk of fault as pump failure or occlusion/disconnection prevents the formula from being infused, leading to severe hypoglycaemia, seizures or death [24,44,45,46]. Furthermore, in inexperienced hands, fluctuations in blood glucose following disconnection of the tube in the morning may cause rebound hypoglycaemia [24,25]. Waking for night feeds with UCCS is an alternative in children if CNGDF is not possible and may replace CNGDF in adults [25]. In summary, multiple factors can determine the mode of nocturnal dietary treatment, and it is important to note that missing one bolus feed can result in severe hypoglycaemia with lactic acidosis, seizures and death. The chosen approach to nocturnal dietary management should be a shared decision between families and healthcare providers, following careful discussion of the pros and cons of each approach and taking into account individual preferences and circumstances [47]. In practice, many families use UCCS and CNGDF depending on what is most convenient or appropriate at a given time based on their resources. Given the extensive home management of the disease, acute complications encountered at home or resulting in hospitalisation in patients with GSDIa are presumably underreported irrespective of the mode of dietary management.

When introducing UCCS, it should be done gradually to improve tolerance [24]. Dietary treatment with UCCS should be individualised in terms of dose and dosing interval; however, general guidelines for dosing are 1–1.6 g per kg (ideal body weight) every 3–4 h for young children (although children under 3 years of age may require more frequent treatment) and 1.7–2.5 g per kg (ideal body weight) every 4–5 h for older children, adolescents and adults [16,24]. Dosing may vary between countries depending on the type and/or quality of the UCCS used (e.g., purity, amylopectin content). The dose may need to be adjusted during periods of increased metabolic demand, such as during exercise, puberty and pregnancy, to avoid undertreatment and the risk of hypoglycaemia [48]. Cornstarch requirements decrease with age, and the dose may need to be reduced in adults with GSDIa [49]. Failure to decrease the dose may result in overtreatment, which can lead to obesity, hyperlipidaemia and metabolic instability, worsening hepatomegaly, hyperinsulinaemia and rebound hypoglycaemia [48,49]. However, there are no robust evidence-based guidelines to support age-related dosing or adjustments of the dosing interval.

Heat modified waxy maize cornstarch preparations have shown improvements in the duration of normoglycaemia and glucose profile compared with UCCS in some patients with GSDIa [50,51]. An emulsifiable version of this heat modified waxy maize cornstarch (Glycosade^®^, Vitaflo International, Liverpool, UK) is available in the USA, North America, Europe, Australia, China, the Middle East and parts of Latin America; however, cost means access is limited in many countries, such as Brazil and Colombia. Glycosade^®^ has shown some benefits compared with traditional UCCS; in a clinical trial of adults and children with GSDIa, 82 of 93 (88%) patients maintained glucose concentrations above 70 mg/dL for 2 h longer with Glycosade^®^ compared with UCCS following an overnight challenge [52]. Glycosade^®^ is not approved for nocturnal use in children under 5 years of age in the USA [48]. Sweet manioc starch (SMS) is extracted from cassava (*Manihot esculanta*), which is a staple food crop in many developing countries. In in vitro analyses, SMS demonstrated slower glucose release compared with other starches (including UCCS and Glycosade^®^) [53]. A randomised, triple-blind, Phase I/II, crossover pilot study was conducted in 11 patients (aged ≥ 16 years) with GSDIa and found SMS was non-inferior to UCCS for the maintenance of euglycaemia [54]. All patients had evidence of poor metabolic control and experienced an increase in lactate levels >2.2 mmol/L, which warrants further investigation; however, the wide availability and relatively low cost may mean SMS is a useful product for preventing GSDIa-related hypoglycaemia [54].

UCCS therapy, along with strict nutritional management and monitoring of multiple biomedical parameters, is the current standard of care for GSDIa [24,25]. Blood glucose monitoring most often takes the form of capillary blood glucose tests (self-monitoring of blood drop samples, usually collected from fingertip pricks [55], 3–6 times per day) or 24-h continuous glucose monitoring (CGM) via a disposable sensor placed on the back of the arm or abdomen. However, CGM may be limited by regional availability and/or reimbursement by insurance or government-funded health care [55,56]. Blood glucose monitoring should be performed to establish the initial diet prescription. A dietitian specialising in metabolic diseases is integral to optimising the diet, ensuring small frequent feedings that are high in complex carbohydrates are spread evenly throughout the day (when required). The nutrient composition of the diet should be broken down as follows: 60–70% of calories to be obtained from carbohydrates (food and UCCS), 10–15% of calories to be obtained from protein (to provide the daily recommended intake) and the remaining calories to be obtained from fat (<30% for children older than 2 years) [24]. Simple sugars (fructose, sucrose and galactose) should be restricted as they are not metabolised beyond G6P and may contribute to lactic acidosis, glycogen accumulation and rapid insulin secretion [57], although no consensus exists regarding the extent of this advice. Sugar alcohols, especially sorbitol, should also be avoided. The restricted nature of the diet means supplementation is necessary to meet the recommended dietary allowance (RDA) and avoid deficiencies in micronutrients; individuals with GSDIa require a sugar-free multivitamin, calcium, fish oil and vitamin D supplementation. Supplementation to meet the RDA and avoid deficiencies in macronutrients may also be required. Furthermore, some individuals require the addition of non-carbohydrate calorie sources in the form of protein and fat to meet estimated calorie needs. Patients with GSDIa are at risk for several of the top five ranked criteria for malnutrition diagnosed by the 2016 Global Leadership Initiative on Malnutrition (GLIM) [58]. These include phenotypic criteria, namely reduced muscle mass, weight loss and low body mass index (BMI), as well as aetiologic criteria, specifically reduced food intake and disease burden/inflammation. Malnutrition is diagnosed when at least one phenotypic criterion and one aetiologic criterion are identified. The most commonly seen phenotypic criterion observed in individuals with GSDIa is reduced muscle mass; consequently, many require protein supplementation. Hepatic glycogen storage diseases are associated with microbial dysbiosis [59]; therefore, supplementation with probiotics is recommended. Alcohol should be restricted because of the risk of liver inflammation and aberrant alcohol metabolism [48]. Depending on the individual patient’s metabolic control, blood glucose monitoring should also be performed before meals or UCCS intake, as well as before and after exercise, in times of stress/illness or when low blood glucose is suspected [24].

Additional treatments or interventions may be required to treat or manage long-term complications and comorbidities. Typically patients with GSDIa may receive the following medications on a daily basis: xanthine oxidase inhibitors to prevent gout (if blood uric acid concentration is not normalised by dietary treatment), angiotensin-converting enzyme inhibitors or angiotensin-receptor blockers for hyperfiltration and/or albuminuria (even if a patient is normotensive), citrate supplementation for hypocitraturia, lipid-lowering medications (if lipid levels remain elevated despite good metabolic control) and iron for anaemia [24,25].

## 5. Burden of Disease, Management and Monitoring

Patients with GSDIa experience a substantial burden associated with the disease, its management and monitoring and have several unmet biopsychosocial needs (Box 2).

Box 2Unmet biopsychosocial needs and challenges associated with Glycogen Storage Disease typle Ia and its management and monitoring.1.Uncooked cornstarch (UCCS) therapy does not correct the underlying cause of disease, and patients remain at risk of serious long-term complications (e.g., hepatocellular adenomas, hepatocellular carcinoma, renal disease, osteoporosis) [24,42].2.Some long-term complications develop even with good glycaemic and metabolic control [1,20,24].3.Patients may experience hypoglycaemia despite adhering to dietary treatment [20].4.Constant home/self-monitoring of glucose levels and adjustment of dosing is required to reflect changing metabolic needs and prevent overtreatment or undertreatment [24,60].5.The therapeutic range between dietary under treatment and overtreatment is complicated by imprecision in dosing of UCCS, which can lead to fluctuating glucose levels [48].6.Undertreatment increases the risk of hypoglycaemia and consequent serious long-term complications; overtreatment can lead to weight gain, hyperlipidaemia and metabolic instability, worsening hepatomegaly, hyperinsulinaemia and rebound hypoglycaemia [49,60].7.Patients may be unaware of hypoglycaemic episodes and remain asymptomatic until blood glucose levels are dangerously low [1,20].8.Self-monitoring of glucose levels may not accurately capture periods of asymptomatic hypoglycaemia, particularly when lactate levels are elevated, and literature supporting the use of continuous glucose monitoring is currently lacking [61].9.Cornstarch is unpalatable and may cause gastrointestinal complaints leading to poor compliance and risk of hypoglycaemia [20,24,50,60].10.Patients may experience reimbursement issues, e.g., for dietary products and/or blood glucose monitoring, including continuous blood glucose monitoring.11.Strict lifelong dietary treatment is challenging to manage and involves avoiding simple sugars and excess carbohydrates; UCCS ingestion requires careful planning around mealtimes [11,24].12.Patients and parents/caregivers suffer disrupted sleep due to the need for night feeds [62].13.Patients and parents/caregivers experience fear, anxiety and stress associated with missed doses of UCCS and whether they might prove fatal [62].14.Fear of hypoglycaemia may lead to overfeeding, which can cause obesity [60,63].15.Patients may have concerns or experience low self-esteem related to their physical appearance [64].16.Patients may experience disrupted schooling due to time off/hospitalisation relating to the disease, which may impact future career choices [62].

### 5.1. Burden of Disease

The risk and consequences of hypoglycaemia are constant; individuals with GSDIa must always carry UCCS preparations with them as standard and may require extra for use in emergencies. Missed doses, e.g., due to failure to wake for a night feed, can lead to hypoglycaemia, seizures and death [24]. Patients may experience a hypoglycaemic event despite adhering to their UCCS dosing schedule; in a retrospective study, one-third of patients experienced episodes of coma despite treatment [20]. Recurrent hypoglycaemia can blunt the counter-regulatory responses responsible for preventing hypoglycaemia; this hypoglycaemia unawareness may result from an altered ability of the brain to sense symptoms of neuroglycopenia and/or impaired coordination of the counter-regulatory responses themselves [60]. The brain uses lactate as an alternative energy source resulting in the absence of cerebral symptoms despite severe hypoglycaemia [65]. Consequently, patients may be unaware of their hypoglycaemia and remain asymptomatic until blood glucose levels are dangerously low [1,20]. Patients with low glucose levels tend to have high lactate concentrations in the blood [65]. Lactate concentrations are not monitored routinely and can rise rapidly in times of intercurrent illness, leading to lactic acidosis [1]. Lactate concentrations start to rise prior to hypoglycaemia and may blunt fasting-associated symptomatology. Notably, patients with GSDIa who have some G6Pase activity or borderline glucose control may have chronically elevated lactate levels, which has been associated with long-term complications [1]. Fear of hypoglycaemia can result in excessive carbohydrate intake. Excess carbohydrate will also lead to glycogen accumulation in the liver and result in progressive hepatomegaly; it can also lead to lactic acidosis. Timing of UCCS ingestion also needs to be carefully planned around mealtimes as it can lead to reduced appetite and ultimately anorexia, which coupled with the restricted diet can result in nutritional deficiencies [11].

### 5.2. Burden of Disease Management and Monitoring

Ingestion of UCCS itself can be burdensome, as the limited duration of action means large volumes must be carried at all times to allow small doses to be consumed every few hours according to the individual patient’s needs. For example, for a 10-day trip, an individual may need to take 3–4 kg of UCCS plus the equipment required for preparation, administration and monitoring. Extended-release formulations (Glycosade^®^) are available, mainly for overnight management, but still require frequent ingestion [11,24,62]. UCCS is unpalatable and may cause gastrointestinal complaints, including bloating, flatulence and diarrhoea, leading to poor compliance [20,24,48,50]. Glucose replacement therapy with UCCS can be imprecise; in many centres and homes, it is dosed by volume (e.g., tablespoons and scoops), which can lead to wide fluctuations in glucose levels and periods of hyperglycaemia and hypoglycaemia [48,49]. Accuracy of dosing may be improved by weighing UCCS during preparation; however, the composition of the daily diet can influence the absorption of UCCS, and requirements may change depending on daily activity levels.

Strict home/self-monitoring of blood glucose levels and adjustment of UCCS dosing is required to ensure appropriate glycaemic control for daily activities, e.g., when participating in sports or other physical activity or in times of acute illness [48]. CGM is an increasingly accurate and powerful tool for monitoring daily fluctuations in blood glucose levels widely used in patients with type I diabetes [49,66]. Several published studies have assessed the utility of CGM in a number of GSDs, including a clinical trial of patients with GSD type I [56,61,66,67,68,69]. Of note, studies performed pre-2015 used first- or second-generation CGM devices; these were less accurate than newer models, and results should be interpreted with caution. Measurement errors and physiological differences between capillary blood and interstitial fluid glucose levels affect CGM accuracy, and both prospective and validation studies must be performed to determine reference values for CGM outcome parameters and to correct for intra-patient variation [56]. The amount of subcutaneous fat affects the accuracy of CGM, which is an important consideration when monitoring infants and patients with increased BMI [56]. CGM may alleviate some of the burden of glucose monitoring by fingertip prick tests, but these remain the standard approach in many countries at present.

Maintaining good glycaemic and metabolic control is crucial as it correlates with long-term outcomes; however, the lack of precision resulting in fluctuating glucose levels and potential for overtreatment and undertreatment within individual patients means this is difficult to achieve with UCCS therapy and may contribute to the risk of metabolic syndrome [70]. This represents a substantial unmet need in patients with GSDIa. Individual risk of long-term complications is not well understood and is likely related to multiple factors, including age at diagnosis and when treatment is started. Extended periods of poor metabolic control may explain the increased susceptibility to HCC in patients homozygous for the c.648G>T pathogenic variant who tend to be diagnosed in adulthood, as already mentioned, and Wang et al. demonstrated a link between poor metabolic control (based on prolonged serum triglyceride levels >500 mg/dL) and HCA progression [27]. Poor metabolic control may contribute to bone disease through mechanisms of hypoglycaemia counter-regulation [71]. Glucagon produced in response to hypoglycaemia activates cAMP formation in the liver through G-protein coupled receptors resulting in increased cellular G6P. G6P cannot be converted to glucose but can be converted to pyruvate and reduced to lactate, or it can undergo oxidative decarboxylation and be converted to acetyl-coenzyme A. Acetyl-coenzyme A can shunt into the lipid biosynthesis pathway to produce fatty acids. Shunting G6P down the glycolytic pathway leads to lactic acidosis (Figure 1) and causes bone mineral loss by inducing bone mineral dissolution, enhancing bone resorption by osteoclasts and inhibiting bone formation by osteoblasts [72]. Preclinical models have shown excess glucocorticoids that also affect bone formation and resorption processes by reducing osteoblast production and lifespan and prolonging the lifespan of osteoclasts [73]. Serum cortisol levels are higher in individuals with GSDIa versus age- and sex-matched controls under normoglycaemic circumstances [19]. The increased cortisol concentration may be due to local deregulation as opposed to hypothalamo–pituitary–adrenal activation [19]. Although acute hypoglycaemia leads to increased cortisol concentrations, cortisol homeostasis is not well characterised in hypoglycaemic patients with GSDIa.

Finally, from a healthcare provider perspective, in the event of hospitalisation due to metabolic decompensation, local or regional healthcare providers are often responsible for initiating emergency treatment; however, many feel they have insufficient knowledge to start this independently. Prompt safe and effective emergency treatment of metabolic decompensation is crucial for patients with GSDIa, and an international effort has been made to develop a simple, shared emergency protocol, which can be generated online at any time by local physicians and families [74].

## 6. Quality of Life

GSDIa is an inherited, chronic and incurable disease that requires constant monitoring of blood glucose levels and strict dietary treatment to avoid hypoglycaemia. It is also associated with serious long-term complications and comorbidities. All of these aspects can have a negative impact on QoL and health-related QoL in adult and paediatric populations from both the patient and parent/caregiver perspective [41,75].

Eating disorders in children and adults with GSDIa [64,76] and the psychosocial impact of the disease in adults [77] have been reviewed. Additionally, parents/caregivers of children with GSDIa experience greater stress and distress versus parents/caregivers of children without a chronic disease [41]. This may be due to the burden and/or complications associated with enteral feeding; for example, PEG-/G-tubes are associated with recurrent infection at the surgical site and may require treatment/hospitalisation. In some cases, further surgery may be required to close the surgical site upon removal of the PEG/G-tubes [78]. Additionally, pumps may fail or become occluded or disconnected [24]. Parents/caregivers may experience fear, anxiety and stress associated with missed doses of UCCS and the irreversible consequences this may have on the child [62], as well as interrupted sleep due to the need to administer night feeds [79]. Parents/caregivers may also have concerns about the need for strict dietary treatment and frequent ingestion of UCCS in large volumes and the impact this has on the child (e.g., side effects such as stomach ache, constipation, diarrhoea, feeling self-conscious taking it at school or in front of friends) [62]. Other worries include fear of hospitalisation following gastrointestinal disturbances (e.g., vomiting, diarrhoea) that would normally be relatively minor in the general population, disrupted schooling and the impact it may have on future work/career choices, and concerns about growth/height [62]. Families may also experience a direct and/or indirect financial burden associated with caring for a child with GSDIa. Direct costs include UCCS itself as large quantities must be ingested every few hours over the course of the lifetime; costs will also be incurred for treatment of comorbidities and long-term complications. The strict dietary treatment schedule must be adhered to throughout the day and night and may require one parent to give up work to care for the child; such indirect costs may have a significant financial impact on families, particularly single-parent families.

As the child gets older, they are likely to experience a direct impact of the disease on their QoL, which will persist into adulthood. A 2021 study of adults (aged 17–54 years) with type I GSD (27 with GSDIa; 7 with GSDIb) published by Garbade et al. assessed the impact of the disease on daily life and reported that most patients considered their disease to be one of moderate severity and disease burden that can be challenging to manage with dietary treatment [77]. Patients with GSDIa have reported reduced physical, social and psychosocial functioning versus healthy control populations [20,41,75]. Interrupted sleep due to the need for night feeds [62] or sleep disturbances [79] associated with unnoticed hypoglycaemia is a major issue for many patients. Strict dietary treatment, the need to ingest UCCS frequently and the inability to leave home for extended periods without taking large volumes of UCCS may result in difficulties socialising or participating in school/work, leisure and physical activities. Feeling different from their peers is an issue for children and teenagers, who may have to interrupt the school day to ingest UCCS [79]. Not being able to have a normal social life has been highlighted by patients as one of the most frustrating aspects of the disease; dietary restrictions mean activities such as eating out prove challenging. The need to consume UCCS to prevent hypoglycaemia has impacted patients’ willingness to exercise [62], and Garbade et al. found that more than 85% of patients in their study considered their physical fitness to be moderately or severely affected by their disease [77].

Garbade et al. also reported that the three negative emotions most commonly felt by patients in relation to their disease were anxiety, fear and rage [77]. Individuals may experience fear, anxiety and stress associated with missed doses of UCCS and the potential consequences, as well as concerns regarding physical appearance (e.g., due to growth retardation, weight gain or body shape) [64,79]. Weight gain has been highlighted by patients as another of the most frustrating aspects of the disease [62], and patients are at risk of developing eating disorders, such as avoidant/restrictive food intake disorder [64,76]. A large proportion of daily calories are obtained from UCCS, which may affect appetite, and chronic use of continuous overnight enteral feeding has been associated with eating disorders and anorexia. Feeding difficulties and orofacial myofunctional disorders are prevalent in patients with GSD, including GSDIa, as are disordered eating habits and decreased smell and taste perception [80]. Poor mental health in general may be an issue as the disease impacts every aspect of patients’ lives [62].

QoL may be further affected by long-term complications and comorbidities of GSDIa, including obesity and metabolic syndrome as a consequence of the carbohydrate-rich diet and/or UCCS; fear of hypoglycaemia may cause patients to overeat or consume excess UCCS contributing to obesity [63]. Obesity, in turn, increases the risk of dyslipidaemia, although premature atherosclerosis is rare [81]. Other long-term complications that may impact QoL include HCA and the associated risk of bleeding or developing HCC, renal disease and the potential for renal failure [42], osteoporosis and fractures resulting from vitamin D deficiency, chronic lactic acidosis and PCOS and potential fertility issues in females [16].

## 7. Future Directions

GSDIa can be a lethal disease without dietary treatment to prevent hypoglycaemia and secondary metabolic perturbations [10]. As survival has improved, so has the understanding of the complex biological, medical and psychosocial needs of individuals with GSDIa and their families as they progress into adolescence and adulthood. A multidisciplinary team approach is required to develop a detailed care plan outlining the dietary treatment and drug therapy required to manage the disease and associated long-term complications that is tailored to the individual needs of each patient, taking into account their QoL. Education of the patient and their family is essential to ensure they have the necessary knowledge and motivation to cope with the responsibility of managing their treatment and care as they become more independent. Additional support may be required to help the patient and their family handle the psychological burden of assuming or passing on this responsibility at what may be a particularly anxious time for those involved.

If euglycaemia is maintained, metabolic perturbations and clinical parameters can be improved in many patients; however, they are not fully corrected nor is the underlying cause of the liver and kidney disease. Even if patients adhere to strict dietary treatment, complications, such as hyperlipidaemia, hyperuricaemia, lactic acidosis, HCA, HCC and renal disease, cannot always be prevented [10]. HCA with the potential for transformation into HCC remains one of the most concerning long-term complications of GSDIa [24]. Boers et al. published a retrospective, observational study of 80 patients with GSD type I, in whom liver transplantation was indicated because of HCA/liver abnormalities/focal nodular hyperplasia (29 patients), poor metabolic control (27 patients), growth retardation (13 patients, some with delayed puberty and sexual maturation), renal failure (five patients, three of whom received a combined liver and kidney transplant), bleeding complications leading to anaemia (one patient) and acute pancreatitis due to severe hypertriglyceridaemia (one patient) [82]. Patients with GSDIa who undergo liver transplantation are at risk of severe lactic acidosis in the perioperative period if managed by healthcare providers with limited experience in GSDIa [83]. While patients with GSD type I who receive a functional liver transplant achieve normal metabolic control and normal fasting tolerance, they are at risk of complications associated with the transplant itself and require subsequent immune suppression [82]. Normalisation of metabolic control has also been achieved with human hepatocyte transplantations; however, the beneficial effect was not sustained long term [82,84,85]. Liver transplantation does not protect against renal complications; therefore, a combined liver and kidney transplant may be considered. Decisions regarding when to perform liver transplants are complex and should be made at the individual level, which requires a thorough multidisciplinary understanding of the risks and benefits associated with transplantation, as well as the disease, dietary treatment and any forthcoming novel therapeutic alternatives [24,82,86]. The development of new therapeutic opportunities for GSDIa, such as gene therapy using AAV8 delivery vectors that are being tested in ongoing clinical trials (NCT03517085 and NCT03970278) [87], might offer the potential to relieve the burden associated with strict dietary treatment, improve glycaemic control and prevent long-term complications that arise due to recurrent hypoglycaemia and related biochemical abnormalities, thus improving QoL of those affected. Other therapeutic approaches in preclinical development include mRNA therapy, gene editing, AAV9 gene therapy and exon skipping, and clinical data are awaited.

Integration of care and research, monitoring from home and standard outcome measures are becoming increasingly important to characterise rare diseases [88]. Research priorities have been formulated for liver GSD in an international priority-setting partnership [89]. Identification of the best methods for ensuring sufficient amounts of fully functional enzyme in patients with liver GSD was ranked as a top priority overall and specifically for patients with GSDIa. The importance of QoL as an overarching priority, as opposed to a single priority, in patients with liver GSD was emphasised. Self-monitoring and treatment are important for patients with GSDIa because they receive immediate physical feedback (by becoming hypoglycaemic) if dietary treatment is failing. Current guidelines [24,25] and care pathways have been developed relating to patients with GSDIa instead of in collaboration with them. In future, patients and their families should be involved in the selection of outcome measures that matter most to them. Important lessons can be learned from another, more prevalent disorder in which glucose homeostasis is perturbed. Recently Nano and co-workers reported on a standard set of outcome measures for diabetes mellitus in collaboration with the International Consortium for Health Outcome Measurement (ICHOM) [90]. Standard outcome sets, such as those developed by the COMET initiative, are not only important tools to deliver value-based healthcare, as a standard set of patient-reported outcome measures specific for patients with GSDIa may become an important instrument in the efficacy assessment of novel, innovative therapies that are under investigation.

## 8. Conclusions

Oral glucose replacement therapy in the form of UCCS and strict control of nutritional treatment has significantly improved the prognosis of patients with GSDIa. However, achieving optimal glycaemic and metabolic control is challenging, and the risk of severe hypoglycaemia and chronic complications persists. Both the disease itself and the need for meticulous adherence to dietary treatment have a significant physical, psychological and psychosocial burden on patients and their parents/caregivers. There is an unmet need for novel monitoring tools that measure outcomes that matter to patients and for treatment strategies that address the underlying cause of disease, restore glucose homeostasis, prevent secondary metabolic perturbations and chronic complications, and improve QoL.

## Figures and Tables

**Figure 1 nutrients-13-03828-f001:**
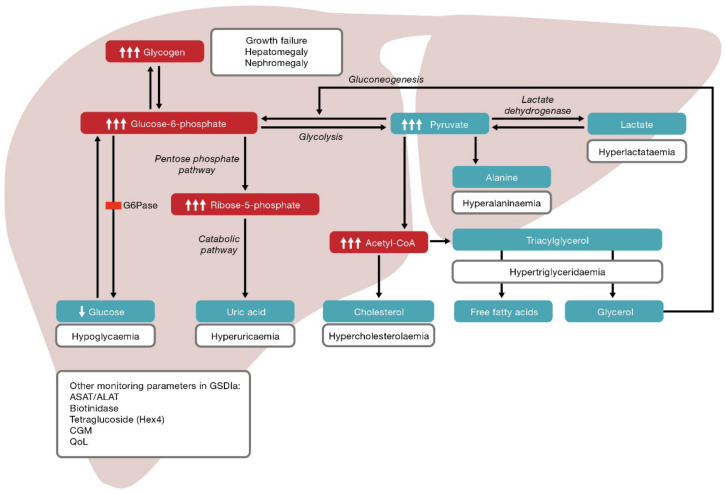
Consequences of defective G6Pase. Defective G6Pase means glucose-6-phosphate accumulates and is shunted into other pathways leading to hyperuricaemia, hypercholesterolaemia, hypertriglyceridaemia, hyperalaninaemia and hyperlactataemia, in addition to hypoglycaemia. Acetyl-CoA—acetyl-coenzyme A; ASAT—aspartate transaminase; ALAT—alanine transaminase; CGM—continuous glucose monitoring; G6Pase—glucose-6-phophatase; QoL—quality of life.

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
