# Peer review of "Glycogen Storage Disease Type Ia: Current Management Options, Burden and Unmet Needs"

_nutrients, 2021, doi:10.3390/nu13113828_

Round 1
Reviewer 1 Report
In this review manuscript, Derks et al. illustrated glycogen storage disease type Ia from epidemiology to the view of quality of life. The manuscript was well-written, and provided a lot of important information for the readers. However, several issues are still needed to clarified and revised.
- Although the main concept is Glycogen storage disease type Ia, another types of glycogen storage disease are suggested to make a brief introduction and provide in the first paragraph of the manuscript.
- In lane 84-85, the authors indicated that there are a lot of factors increased blood glucose levels, but only insulin decreased blood glucose levels. Since the role of dysbiosis is important in glycogen storage disease type Ia, and glucagon like peptide-1 is another factor that decreases blood glucose, glucagon like peptide-1 is suggested to replenish in lane 84-85.
- In Current strategies for disease management section, the possible mechanisms for each diet for the treatment of glycogen storage disease are encouraged to provide in this section. This would be helpful for researchers to develop novel therapeutic strategies for the treatment of glycogen storage disease.
Reviewer 2 Report
good work, great informations
Reviewer 3 Report
I read with great interest your manuscript on GSD Ia wherein you describe the current management options with focus on the burden for patients and caregivers and the unmet needs. The epidemiology; pathophysiology and clinical manifestations of this rare disease are very well described in a comprehensive manner. The current strategy for disease management is described in detail, however I miss an overview of the therapeutic targets: can they be summarized in a table or box as you have done for the unmet needs and burden, encompassing biological (BMI) and biochemical targets (glycemia; lactic acid; serum triglycerides and total cholesterol,...) as this will be very helpful to caregivers, esp. specialized dietitians. You mentioned (p5) vitamin B deficiency as one of the nutritional deficiencies: is this focusing on thiamine deficiency in particular? Liver steathosis and adenomas are mentioned in the text: when and how frequent should the liver be examined for these complications and how (MRI?; ultrasonography?)? Future directions are promising: does gene therapy also prevent kidney disease that is not achieved by liver transplantation (perhaps this cannot be answered at this moment as the clinical trials are still ongoing).
